# Frequency of Detection and Prevalence Factors Associated with Common Respiratory Pathogens in Equids with Acute Onset of Fever and/or Respiratory Signs (2008–2021)

**DOI:** 10.3390/pathogens11070759

**Published:** 2022-07-02

**Authors:** Nicola Pusterla, Kaitlyn James, Samantha Barnum, Fairfield Bain, D. Craig Barnett, Duane Chappell, Earl Gaughan, Bryant Craig, Chrissie Schneider, Wendy Vaala

**Affiliations:** 1Department of Medicine and Epidemiology, School of Veterinary Medicine, University of California, Davis, CA 95616, USA; smmapes@ucdavis.edu; 2Department of Obstetrics, Gynecology and Reproduction Biology, Massachusetts General Hospital, Boston, MA 02114, USA; kaitlynej@gmail.com; 3Merck Animal Health, Madison, NJ 07940, USA; fairfield.bain@merck.com (F.B.); dcbarnettdvm@gmail.com (D.C.B.); duane.chappell@merck.com (D.C.); emgone@reagan.com (E.G.); bryant.craig@merck.com (B.C.); chrissie.schneider@merck.com (C.S.); wendy.vaala@merck.com (W.V.)

**Keywords:** upper respiratory tract infection, viruses and bacteria, frequency, prevalence factors, qPCR, equids

## Abstract

A voluntary biosurveillance program was established in 2008 in order to determine the shedding frequency and prevalence factors for common respiratory pathogens associated with acute onset of fever and/or respiratory signs in equids from the USA. Over a period of 13 years, a total of 10,296 equids were enrolled in the program and nasal secretions were analyzed for the qPCR detection of equine influenza virus (EIV), equine herpesvirus-1 (EHV-1), EHV-4, equine rhinitis A and B virus (ERVs), and *Streptococcus equi* subspecies *equi* (*S. equi*). Single infections with respiratory pathogens were detected in 21.1% of the submissions with EIV (6.8%) and EHV-4 (6.6%) as the two most prevalent viruses, followed by *S. equi* (4.7%), ERVs (2.3%), and EHV-1 (0.7%). Multiple pathogens were detected in 274 horses (2.7%) and no respiratory pathogens in 7836 horses (76.2%). Specific prevalence factors were determined for each of the six respiratory pathogen groups; most differences were associated with age, breed, and use of the horses, while the clinical signs were fairly consistent between viral and bacterial respiratory infections. Monitoring the frequency of detection of common respiratory pathogens is important in order to gain a better understanding of their epidemiology and to implement management practices aimed at controlling disease spread.

## 1. Introduction

Infectious respiratory diseases are considered one of the most prevalent medical entities in young performance horses and a common reason for athletic horses to be temporarily removed from scheduled training and competitions [1,2,3,4,5]. Various factors have been linked to the development of an upper respiratory tract infection (URTI), including young age, husbandry practice of congregating large groups of horses, management practices (shared use of tack and other equipment), suboptimal biosecurity protocols, environmental conditions (exposure to dust and chemical irritant such as ammonia), immune-suppression due to exercise and transportation, and various levels of protection secondary to routine vaccination [6,7].

Equine influenza virus (EIV), equine herpesvirus-1 (EHV-1), EHV-4, equine rhinitis A (ERAV) and B (ERBV) virus (ERVs), and *Streptococcus equi* subspecies *equi* (*S. equi*) are considered common respiratory pathogens associated with URTI. The reported frequency of detection of these pathogens varies with the population of horses studied, geographic origin of the studied population, time of the year, as well as laboratory tests used to determine infection [8,9,10,11,12,13].

With the increase in national and international movement of equids for breeding and sporting events, the likelihood of respiratory pathogen transmission has increased, as evidenced by recent outbreaks worldwide [14,15,16,17,18,19]. The investigation of risk factors for the development of URTI and determining specific factors for each respiratory pathogen can shed light on their epidemiology and help implement practices aimed at controlling disease spread. It was therefore the aim of this study to describe the detection frequency of respiratory pathogens and selected prevalence factors in equids with acute onset of fever and/or respiratory signs enrolled in an industry-sponsored voluntary surveillance program from 2008 to 2021.

## 2. Results

The study population was composed of 10,296 equids with acute onset of fever and/or respiratory signs for which rostral nasal secretions were submitted for the testing of selected respiratory pathogens. The number of submissions per year ranged from 325 to 1058 (median 780 submissions per year). Submissions from index cases were similar in numbers per meteorological season (Table 1). Demographically, the population was composed of a variety of breeds with Quarter Horse being overrepresented followed by other breeds, Thoroughbred, and Warmblood (Table 2). Sex was equally distributed within the study population while the age groups 1–4 and 4–9 years of age were the most common age groups represented. The majority of the study horses were used for competition and pleasure. There was no history of recent transportation in 64.1% of the cases and 65.7% of the index cases were reported as single cases. The most frequently reported clinical signs in the entire study population were, in decreasing frequency, fever (73.0%), nasal discharge (67.8%), lethargy (65.3%), anorexia (53.8%), and coughing (43.5%; Table 3). Approximately one-third of all index cases had reported vaccination history against EIV and/or EHV-1/-4, while vaccination against *S. equi* was seldom reported. Single infections with respiratory pathogens were detected in 21.1% of the submissions with EIV (6.8%) and EHV-4 (6.6%) as the two most prevalent viruses, followed by *S. equi* (4.7%), ERVs (2.3%) and EHV-1 (0.7%). Multiple pathogens were detected in 274 horses (2.7%) and no respiratory pathogens were detected in 7836 horses (76.2%). Co-infections involved any combination of two and three of the tested pathogens, with EIV and EHV-4, EHV-4 and *S. equi,* and *S. equi* and ERVs representing the three most common combinations (Table 4).

When EIV qPCR-positive horses were compared against the entire cohort of horses with known respiratory infections, the following prevalence factors were determined: (i) horses were more likely to test EIV qPCR-positive in the spring and less likely in the fall (*p* < 0.001; Table 1); (ii) EIV qPCR-positive horses were seen more frequently in the 5–9 years of age group (*p* < 0.001; Table 2); (iii) EIV qPCR-positive horses were more prevalent in “other breed” and less prevalent in Thoroughbred (*p* < 0.001); (iv) EIV qPCR-positive horses were more prevalent in pleasure use and less commonly detected in competition use (*p* < 0.001); (v) EIV qPCR-positive horses were associated with recent transportation history and multiple affected horses on the property (*p* < 0.001). There was no difference in the frequency and level of fever between EIV qPCR-positive horses compared to horses with other respiratory pathogens (*p* > 0.05; Table 3). EIV qPCR-positive horses were more likely to display nasal discharge (*p* < 0.001) and coughing (*p* = 0.001) compared to the horses with other infectious etiologies. There were no statistical differences in lethargy and anorexia between EIV qPCR-positive horses and equid with other respiratory infections (*p* > 0.05).

When EHV-4 qPCR-positive cases were compared against the entire cohort of horses with known respiratory infections, the following prevalence factors were determined: (i) EHV-4 qPCR-positive cases were more likely in the fall and less likely in the spring (*p* < 0.001; Table 1); (ii) the frequency of EHV-4 qPCR-positive cases was the highest in the < 1 year of age group (Table 2); (iii) EHV-4 qPCR-positive cases were seen with greater detection rate in Thoroughbreds and less frequently in ponies (*p* < 0.001); (iv) EHV-4 qPCR-positive cases were more prevalent in competition use; (v) EHV-4 qPCR-positive cases were associated with no history of transportation (*p* = 0.004) and single affected horses on the property (*p* = 0.006). Amongst clinical signs, EHV-4 qPCR-positive cases displayed slightly higher rectal temperature (*p* = 0.01), and lower reported detection of nasal discharge, and coughing (*p* < 0.001; Table 3). There were no statistical differences in lethargy and anorexia between EHV-4 qPCR-positive horses and equids with other respiratory infections (*p* > 0.05).

When comparing *S. equi* qPCR-positive cases with cases qPCR-positive for all the other tested pathogens, the following prevalence factors were determined: (i) *S. equi* cases were less likely in the fall (*p* < 0.001; Table 1); (ii) *S. equi* qPCR-positive cases were more likely in horses older than 1 year of age (*p* < 0.001) and in Quarter horses and ponies (*p* < 0.001; Table 2); (iii) *S. equi* cases were seen with greater frequency in pleasure horses and with less frequency in horses used for competition (*p* < 0.001); (iv) there was no association between *S. equi* qPCR-positive cases with number of affected horses on the premise and history of recent transportation (*p* > 0.05). Amongst clinical signs, *S. equi* qPCR-positive cases displayed higher reported detection of nasal discharge (*p* < 0.001) and anorexia (*p* < 0.001), and lower frequency of coughing (*p* < 0.001; Table 3). There were no statistical differences in rectal temperature and lethargy between *S. equi* qPCR-positive horses and equids with other respiratory infections (*p* > 0.05).

Because of the small numbers of ERAV qPCR-positive cases, ERAV and ERBV qPCR-positive cases were combined and reported as ERVs qPCR-positive cases. When the qPCR-positive ERVs cases were compared to the cases qPCR-positive for all other respiratory pathogens, the following prevalence factors were determined: (i) there were no differences in seasonality (Table 1); (ii) ERVs qPCR-positive cases were more prevalent in equids < 1 year of age (*p* < 0.001; Table 2); (iii) ERVs qPCR-positive cases were seen with greater frequency in Thoroughbreds (*p* = 0.02) and in horses used for competition; (iv) ERVs qPCR-positive cases were less commonly associated with the history of transportation (*p* = 0.002) and with single affected horses on the property (*p* < 0.001). There were no statistical differences in rectal temperature, anorexia, and lethargy between ERVs qPCR-positive horses and equids with other respiratory infections (*p* > 0.05; Table 3). Nasal discharge and cough were less frequently reported in ERVs qPCR-positive horses when compared to cases infected with other respiratory pathogens (*p* < 0.001).

Amongst the 78 EHV-1 qPCR-positive cases, the following prevalence factors were determined: (i) index cases were more likely to test EHV-1 qPCR-positive in the spring and less likely in the summer/fall compared to horses infected with all the other pathogens (*p* < 0.001; Table 1); (ii) EHV-1 qPCR-positive horses were more frequently detected in the age groups 1–4 and 10–14 years of age (*p* = 0.001; Table 2); (iii) Thoroughbreds and Warmbloods were more frequently infected with EHV-1 compared to all the other respiratory pathogens (*p* < 0.001); (iv) EHV-1 qPCR-positive cases were more prevalent in competition use (*p* < 0.001); (v) number of affected horses on the premise and recent transportation showed no statistical difference between EHV-1 qPCR-positive horses and horses infected with all other respiratory pathogens (*p* > 0.05). There were no statistical differences in rectal temperature and lethargy between EHV-1 qPCR-positive horses and equids with other respiratory infections (*p* > 0.05; Table 3). Anorexia, nasal discharge, and cough were less frequently reported in EHV-1 qPCR-positive horses when compared to cases infected with other respiratory pathogens (*p* < 0.001).

When horses with multiple infections were compared to the cohort of horses infected with a single pathogen, the following prevalence factors were determined: (i) horses were less likely to be infected with multiple pathogens in the summer compared to horses infected with a single pathogen (*p* < 0.001; Table 1); (ii) age distribution was similar between horses infected with multiple pathogens and horses infected with a single pathogen (Table 2); (iii) Quarter horses were overrepresented in horses infected with multiple pathogens compared to horses infected with a single pathogen (*p* = 0.002); (iv) there were no statistical differences in use, number of horses affected on the premise and recent transportation between horses infected with multiple pathogens, and horses infected with a single pathogen. Only the presence of nasal discharge was more frequently reported in horses infected with multiple pathogens compared to horses infected with a single pathogen (*p* < 0.01; Table 3).

Single viral infections were reported in 1696 index cases, while *S. equi* was reported in 490 cases. The two groups were compared between each other to determine possible prevalence factors. *Streptococcus equi* infections were more prevalent in winter and less common in fall compared to viral infections (*p* < 0.001). Horses infected with *S. equi* were less likely to be < 1 year of age (*p* < 0.001) and more likely to be older (age group 5–9; *p* < 0.001) when compared to horses infected with viruses. Bacterial infections were more common in Quarter horses (*p* < 0.001) and less common in Thoroughbreds (0.04) when compared to viral infections. Further, *S. equi* qPCR-positive horses were more likely to be used for pleasure and less likely to be used for competition compared to horses infected with viruses (*p* < 0.001). There were no statistical differences in the number of affected horses, recent transportation history, and frequency of clinical signs between bacterial and viral infections.

## 3. Discussion

The present study showed that viral and bacterial respiratory pathogens were found alone or as co-infections in 23.8% of horses with acute onset of fever and/or respiratory signs. The results are in agreement with previous studies showing that respiratory viruses can be detected from nasal secretions of 16 to 57% of horses with respiratory signs [8,11,12]. Differences in detection rates amongst studies relate to the population of horses tested, the study design (random selection of individual horses versus study focusing on specific outbreaks), the pathogens tested (well-characterized and/or less characterized viruses such as EHV-2 and EHV-5), and testing platforms used (PCR, virus isolation, serology). The authors believe that the present study is unique as it included a very large population of horses across 44 US states and the study focused on a panel of well-characterized respiratory pathogens. While URTI can affect horses of any age, breed, sex, and use, the study data showed that young performance and pleasure horses, and Quarter horses were overrepresented as index cases. Quarter horses represent the highest percentage of resident horses by breed in the US [20]. Further, young age and population dynamics, such as the congregation of large horse groups during equestrian events, are well recognized factors predisposing the transmission of respiratory pathogens [6]. While the study focused on well-characterized viral and bacterial pathogens, yet unknown or poorly investigated viruses could have been responsible for respiratory disease in some index cases. Recent studies focusing on metagenomics analyses have reported on novel equine viruses (picornavirus, protoparvovirus, copiparvovirus) detected in horses with respiratory disease [21,22,23]. Future studies are greatly needed to investigate the epidemiology of novel viruses and their association with clinical disease. While specific prevalence factors were determined for the entire study population, the analysis of such factors is more relevant in the context of individual infections.

Individual infections and co-infections were reported in 21.1% and 2.7% of index cases, respectively. Among the individual infections, EIV and EHV-4 made up 63% of all infections. This is not surprising considering that these two viruses are endemic in most horse populations and commonly associated with outbreaks [19,24,25,26,27,28,29,30]. It is interesting to notice that seasonality, demographics, use, and clinical signs between EIV and EHV-4 qPCR-positive horses differed slightly. While winter months displayed a similar frequency of detection for the various respiratory pathogens, spring months and fall months were predominantly associated with the detection of EIV/EHV-1 and EHV-4 cases, respectively. The reason for differences in seasonal detection is speculative and may relate to age, husbandry, and the use of horses associated with each viral infection. EHV-4 cases were predominantly seen in equids < 1 year of age, which highlights the transmission pattern between mares and foals [31,32]. EIV qPCR-positive cases were seen with greater frequency in young horses ages 5–9 years of age. Recent studies have reported on EIV cases in older and often vaccinated horses, speculating that this observation may relate to vaccine failure [33,34,35,36]. Unfortunately, vaccination history was not reported in close to 60% of the index cases, preventing the assessment between EIV qPCR-positive cases and vaccine status. The observation that EIV cases often originated from premises with multiple sick horses is a reflection of the contagiousness of this virus [29]. Differences between the two main equine viruses were also found in clinical presentation. While both EIV and EHV-4 cases displayed fever, lethargy, anorexia, and nasal discharge, coughing was predominantly associated with EIV qPCR-positive cases. This highlights the pathogenesis of EIV, as this is the only respiratory virus that disrupts superficial layers of the epithelium throughout the respiratory tract [37]. The demographic of ERVs showed similar prevalence factors as EHV-4. ERVs have been shown to mostly infect young performance horses, alone or in combination with other respiratory pathogens [12,38,39,40]. The role of ERVs as respiratory pathogens with clinical implication has repeatedly been documented, despite the fact that these viruses can be found sporadically in healthy horses [40]. The highest frequency of EHV-1 detection was observed during the spring months and affected both young (1–4 years of age) as well as older horses (10–14 years of age). Further, the greater frequency of EHV-1 qPCR-positive results in horses used for competition, emphasizes the risk of this population potentially developing a more severe disease form such as equine herpesvirus-1 myeloencephalopathy (EHM) as reported in recent outbreaks [14,15,41,42]. While EHV-1 infection is routinely supported through the detection of the virus in nasal secretions, the present study did not test concurrent blood to determine viremia. This limitation may have underestimated the true frequency of EHV-1 detection.

*S. equi* infections were the only bacterial infections documented in the present study with a case frequency of 4.7%. Outbreaks due to *S. equi* often affect large breeding operations [43,44]. In the present study, *S. equi* qPCR-positive cases were predominantly seen during the spring/winter months, and in horses older than one year of age, often used for competition and pleasure use. This observation highlights the sporadic nature of *S. equi* infection in older horses and the role of subclinical adult shedders in the transmission of *S. equi*. While sample type has little effect on the detection of respiratory viruses [45,46], it has been shown that deeper respiratory samples are more sensitive for the detection of *S. equi* compared to rostral nasal swabs [47]. It is therefore possible that the detection rate of *S. equi* from nasal swabs is underrepresented in the present study.

Little attention is given to the interpretation of co-infections in the upper respiratory tract of horses. In the present study, equids with dual or triple infections displayed a similar demographic and clinical presentation compared to individual infections. However, horses with co-infections displayed some of the highest frequency in clinical signs including anorexia, fever, nasal discharge, and coughing, suggesting that co-infections may increase the frequency and/or intensity of clinical disease. In calves, clinical signs, especially fever, are often more severe and of longer duration when multiple viruses are involved in respiratory disease infections [48]. It remains to be determined how co-infections with equine respiratory pathogens influence clinical signs, pathogenesis, and shedding. Of interest is the observation of a recent study investigating the effect of EHV-1 and arteritis virus in equine respiratory mucosa explants showing that these two viruses only slightly influenced each other’s infection [49].

Demographic and clinical data comparing equine viral and bacterial URTI are sparse in the veterinary literature. Equine practitioners often struggle to characterize respiratory infections into viral or bacterial, as the treatment differs. The present study data showed subtle differences between the two groups, mostly regarding demographic, and clinical signs. Viral infections were more commonly reported in young performance horses (≤ 4 years of age), while *S. equi* infections were reported with greater frequency in older horses (≥ 5 years of age) used for pleasure. The differences in demographics and use likely relate to susceptibility and spread of the different pathogen groups [50]. Horses infected with *S. equi* displayed a greater frequency of lethargy and anorexia compared to viral infections, which may relate to the magnitude of systemic inflammation triggered by each of these two pathogen groups. A recent study showed that the inflammatory response measured via serum amyloid A was often stronger in horses infected with *S. equi* compared to horses infected with EIV and EHV-4 [51]. From a practical standpoint, the study data do not offer conclusive information aimed at differentiating viral from bacterial URTIs. Concurrent laboratory diagnostic testing is needed to help differentiate between the two infectious respiratory groups.

Study limitations related to the voluntary nature of the study and the lack of randomization of enrolled veterinary clinics. Further, the lack of follow-up samples during the convalescent period and the lack of enrollment of healthy herdmates from the same premise does not establish definitive causality between clinical entity and detection of selected pathogens. Previous epidemiological studies have used additional diagnostic modalities such as viral or bacterial culture and serology to determine the frequency of respiratory pathogens in various horse populations [8,9,10,11,12,13]. It is well established that laboratory-based qPCR is the recommended test for diagnosis of acute cases, while serological tests are important to define epidemiological questions, such as exposure rate [52]. The authors strongly believe that despite the mentioned study limitations, the very large data set encompassing over a decade of sample submissions allowed to determine relevant prevalence factors for selected equine respiratory pathogens.

## 4. Materials and Methods

### 4.1. Study Enrollment, Population, and Sample Collection

The voluntary biosurveillance program for respiratory pathogens was established in March of 2008 and has enrolled 261 equine veterinary practices selected from the client directory of Merck Animal Health. The enrolled practices originated from 44 US regions and showed the following distribution by region: East (37 practices), Midwest (45), South (82), and West (97). Veterinary practices were given collection material (nasal swabs, collection tubes), sample collection instructions, and questionnaires prior to the enrollment into the program. Veterinarians attending the care of a horse with acute onset of fever (T > 101.5 °C) and/or respiratory signs (nasal discharge, cough) were asked to collect nasal secretions using a 6-inch rayon-tipped swab (Puritan^®^ Sterile Rayon Tipped Applicators, Guilford, ME, USA). Case selection and submission occurred from 11 March, 2008 to 30 September, 2021.

Each submission required the completion of a questionnaire, which was filled out by the attending equine practitioner at the time of sample collection. The questionnaire recorded information pertaining to the patient (age, breed, sex), use (racing, show, pleasure, breeding, other), vaccination history, travel history during the past 14 days, number of animals affected on the premise of origin, and presence of clinical signs at the time of sample collection (general attitude, appetite, rectal temperature, nasal discharge, and presence of cough). Samples and questionnaires were sent on ice overnight to the laboratory for next-day processing and analysis. The results for the various qPCR analyses were faxed or emailed to the submitting practice on the same day the samples were received at the laboratory. If samples did not pass quality control, nucleic acid purification of the backup sample and repeated qPCR testing was performed the next day. For samples with persistent inhibition or poor yield of nucleic acids, a new follow-up sample was requested.

### 4.2. Sample Processing and Analysis

Nucleic acid extraction from nasal secretions and whole blood was performed 24 h post-collection using an automated nucleic acid extraction system (QIAcube HT, Qiagen, Valencia, CA, USA) according to the manufacturer’s recommendations. Nasal secretions were assayed for the presence of EIV, EHV-1, EHV-4, and *S. equi* for all submissions using previously reported real-time TaqMan PCR assays [13,53]. Nasal secretions were also tested for ERAV and ERBV starting 1 September, 2012 on a total of 7720 individual submissions using previously reported qPCR assays [40]. Sample quality and efficiency of nucleic acid extraction was determined using a housekeeping gene (equine glyceraldehyde-3-phosphate dehydrogenase (e*GAPDH*)) as previously described [54].

### 4.3. Statistical Analyses

Descriptive analyses (mean, standard deviation, and median) were performed to evaluate the demographic and clinical information from the submission forms. Categorical analyses were performed using a Pearson’s chi-square test to determine the association between observations (age, breed, sex, clinical signs) and each infection group. Each infectious disease group was compared to the other infectious groups. To avoid interpretation bias when multiple pathogens were involved, only horses with a single pathogen were evaluated in each specific group. Horses with co-infections were grouped in a multiple pathogen group. Categorical analyses were also performed between viral (EIV, EHV-1, EHV-4, ERVs) and bacterial infections (*S. equi*). All statistical analyses were performed using commercial software (Stata Statistical Software, Version 14, College Station, TX, USA) and statistical significance was set at *p* < 0.05.

## 5. Conclusions

The present study represents the largest contemporary biosurveillance data set collected from 10,296 equids with acute onset of fever and/or respiratory signs from the US. The study showed that important infectious respiratory pathogens were detected in 23.8% of index cases. The etiology for the remaining 76.2% of animals with clinical signs of URTI was not further investigated. Single infections with respiratory pathogens were detected in 21.1% of the submissions with EIV (6.8%) and EHV-4 (6.6%) as the two most prevalent viruses, followed by *S. equi* (4.7%), ERVs (2.3%), and EHV-1 (0.7%). Multiple pathogens were detected in 274 horses (2.7%) and no respiratory pathogens were detected in 7836 horses (76.2%). The majority of the respiratory pathogens were detected during the colder months of the year, reflecting seasonal differences in population dynamics. In general, viral infections were associated with young performance horses, while infections with *S. equi* were seen in older pleasure horses. Clinical signs were consistent amongst the various infection groups. However, coughing was a clinical hallmark of EIV infection. Monitoring the frequency of detection of common respiratory pathogens is important in order to gain a better understanding of their epidemiology and being able to implement management practices aimed at controlling disease spread.

## Figures and Tables

**Table 1 pathogens-11-00759-t001:** Distribution by meteorological calendar of 10,296 equids with acute onset of fever and/or respiratory signs participating in a voluntary surveillance program for infectious respiratory pathogens.

	Study Population (n = 10,296)	EIV qPCR + Only (n = 699)	EHV-4 qPCR+ Only (n = 677)	*S. equi* qPCR+ Only (n = 490)	ERVs qPCR+ Only (n = 242)	EHV-1 qPCR + Only (n = 78)	Multiple Pathogens qPCR+ (n = 274)	Negative qPCR Results (n = 7836)
**Season** Winter (Dec-Feb) Spring (Mar-May) Summer (June-Aug) Fall (Sep-Nov)	2407 (23.4%) 2853 (27.7%) 2465 (23.9%) 2571 (25.0%)	205 (29.3%) 255 (36.5%) 114 (16.3%) 125 (17.9%)	201 (29.7%) 106 (15.7%) 110 (16.2%) 260 (38.4%)	153 (31.2%) 152 (31.0%) 98 (20.0%) 87 (17.8%)	63 (26.0%) 61 (25.2%) 45 (18.6%) 73 (30.2%)	22 (28.2%) 42 (53.8%) 9 (11.5%) 5 (6.4%)	92 (33.6%) 100 (36.5%) 22 (8%) 60 (21.9%)	1671 (21.3%) 2137 (27.3%) 2067 (26.4%) 1961 (25%)

**Table 2 pathogens-11-00759-t002:** Signalment (age, breed, sex), use, recent travel history, and number of affected animals in 10,296 equids with acute onset of fever and/or respiratory signs participating in a voluntary surveillance program for infectious respiratory pathogens.

	Study Population (n = 10,296)	EIV qPCR + Only (n = 699)	EHV-4 qPCR+ Only (n = 677)	*S. equi* qPCR+ Only (n = 490)	ERVs qPCR+ Only (n = 242)	EHV-1 qPCR + Only (n = 78)	Multiple Pathogens qPCR+ (n = 274)	Negative qPCR Results (n = 7836)
**Age (years)** <1 1–4 5–9 10–14 15–19 ≥20 Not reported	1612 (15.7%) 2536 (24.6%) 2213 (21.5%) 1561 (15.2%) 940 (9.1%) 639 (6.2%) 795 (6.2%)	60 (8.6%) 217 (31.0%) 201 (28.8%) 99 (14.2%) 42 (6.0%) 11 (1.6%) 69 (9.9%)	234 (34.6%) 187 (27.6%) 89 (13.1%) 57 (8.4%) 42 (6.2%) 25 (3.7%) 43 (6.4%)	36 (7.3%) 145 (29.6%) 129 (26.3%) 70 (14.3%) 46 (9.4%) 24 (4.9%) 40 (8.2%)	95 (39.3%) 61 (25.2%) 25 (10.3%) 20 (8.3%) 13 (5.4%) 12 (5.0%) 16 (6.6%)	5 (6.4%) 26 (33.3%) 13 (16.7%) 19 (24.4%) 6 (7.7%) 4 (5.1%) 5 (6.4%)	66 (24.1%) 72 (26.3%) 59 (21.5%) 26 (9.5%) 23 (8.4%) 12 (4.4%) 16 (5.8%)	1116 (14.2%) 1828 (23.3%) 1697 (21.7%) 1270 (16.2%) 768 (9.8%) 551 (7.0%) 606 (7.7%)
**Breed** Quarter horse Thoroughbred Warmblood American Paint Arabian Draft Pony Others	3653 (35.5%) 1606 (15.6%) 1119 (10.9%) 457 (4.4%) 700 (6.8%) 263 (2.6%) 384 (3.7%) 2114 (20.5%)	279 (39.9%) 56 (8.0%) 52 (7.4%) 29 (4.1%) 27 (3.9%) 32 (4.6%) 30 (4.3%) 194 (27.8%)	222 (32.8%) 173 (25.6%) 53 (7.8%) 27 (4.0%) 62 (9.2%) 12 (1.8%) 11 (1.6%) 117 (17.3%)	225 (45.9%) 48 (9.8%) 20 (4.1%) 15 (3.1%) 33 (6.7%) 14 (2.9%) 39 (8.0%) 96 (19.6%)	95 (39.3%) 49 (20.2%) 19 (7.9%) 9 (3.7%) 21 (8.7%) 8 (3.3%) 1 (0.4%) 40 (16.5%)	22 (28.2%) 19 (24.4%) 15 (19.2%) 2 (2.6%) 5 (6.4%) 1 (1.3%) 1 (1.3%) 13 (16.7%)	122 (44.5%) 27 (9.9%) 17 (6.2%) 16 (5.8%) 23 (8.4%) 6 (2.2%) 20 (7.3%) 43 (15.7%)	2688 (34.3%) 1234 (15.7%) 943 (12.0%) 359 (4.6%) 529 (6.8%) 190 (2.4%) 282 (3.6%) 1611 (20.6%)
**Sex** Mare Gelding/Stallion Unknown	3667 (35.6%) 5033 (48.9%) 1596 (15.5%)	273 (39.1%) 320 (45.8%) 106 (15.2%)	278 (41.1%) 306 (45.2%) 93 (13.7%)	171 (34.9%) 257 (52.4%) 62 (12.7%)	90 (37.2%) 120 (49.6%) 32 (13.2%)	22 (28.2%) 44 (56.4%) 12 (15.4%)	95 (34.7%) 140 (51.1%) 39 (14.2%)	2738 (34.9%) 3846 (49.1%) 1252 (16.0%)
**Use** Competition Pleasure Breeding Other Unknown	4170 (40.5%) 3838 (37.3%) 483 (4.7%) 867 (8.4%) 938 (9.1%)	249 (35.6%) 280 (40.1%) 25 (3.6%) 44 (6.3%) 101 (14.4%)	306 (45.2%) 188 (27.8%) 26 (3.8%) 55 (8.1%) 102 (15.1%)	170 (34.7%) 202 (41.2%) 25 (5.1%) 37 (7.6%) 56 (11.4%)	115 (47.5%) 60 (24.8%) 10 (4.1%) 18 (7.4%) 39 (16.1%)	45 (57.7%) 24 (30.8%) 4 (5.1%) 3 (3.8%) 2 (2.6%)	113 (41.2%) 98 (35.8%) 10 (3.6%) 22 (8.0%) 31 (11.3%)	3172 (40.5%) 2986 (38.1%) 383 (4.9%) 688 (8.8%) 607 (7.7%)
**Recent transport** No Yes Unknown	6600 (64.1%) 2733 (26.5%) 963 (9.4%)	374 (53.5%) 268 (38.3%) 57 (8.2%)	450 (66.5%) 179 (26.4%) 48 (7.1%)	301 (61.4%) 152 (31.0%) 37 (7.6%)	171 (70.7%) 50 (20.7%) 21 (8.7%)	52 (66.7%) 24 (30.8%) 2 (2.6%)	157 (57.3%) 84 (30.7%) 33 (12.0%)	5095 (65.0%) 1976 (25.2%) 765 (9.8%)
**Affected horses** Single Multiple Unknown	6766 (65.7%) 2542 (24.7%) 988 (9.6%)	314 (44.9%) 322 (46.1%) 63 (9.0%)	416 (61.4%) 212 (31.3%) 49 (7.2%)	293 (59.8%) 155 (31.6%) 42 (8.6%)	162 (66.9%) 59 (24.4%) 21 (8.7%)	47 (60.3%) 29 (37.2%) 2 (2.6%)	153 (55.8%) 87 (31.8%) 34 (12.4%)	5381 (68.7%) 1678 (21.4%) 777 (9.9%)

**Table 3 pathogens-11-00759-t003:** Clinical signs and vaccination history in 10,296 equids with acute onset of fever and/or respiratory signs participating in a voluntary surveillance program for infectious respiratory pathogens.

	Study Population (n = 10,296)	EIV qPCR + Only (n = 699)	EHV-4 qPCR+ Only (n = 677)	*S. equi* qPCR+ Only (n = 490)	ERVs qPCR+ Only (n = 242)	EHV-1 qPCR + Only (n = 78)	Multiple Pathogens qPCR+ (n = 274)	Negative qPCR Results (n = 7836)
**Clinical signs** Lethargy yes Lethargy no Lethargy unknown Anorexia yes Anorexia no Anorexia unknown Fever yes Fever no Fever unknown Median temperature Nasal dis. yes Nasal dis. no Nasal dis. unknown Coughing yes Coughing no Coughing unknown	6727 (65.3%) 3173 (30.8%) 396 (3.8%) 5539 (53.8%) 4348 (42.2%) 409 (4.0%) 7520 (73.0%) 1894 (18.4%) 882 (8.6%) 102.9 °F 6980 (67.8%) 2975 (28.9%) 341 (3.3%) 4481 (43.5%) 5401 (52.5%) 414 (4.0%)	506 (72.4%) 158 (22.6%) 35 (5.0%) 395 (56.5%) 272 (38.9%) 32 (4.6%) 609 (87.1%) 47 (6.7%) 43 (6.2%) 103.0 °F 625 (89.4%) 46 (6.6%) 28 (4.0%) 589 (84.3%) 74 (10.6%) 36 (5.2%)	464 (68.5%) 190 (28.1%) 23 (3.4%) 372 (54.9%) 282 (41.7%) 23 (3.4%) 600 (88.6%) 53 (7.8%) 24 (3.5%) 103.0 °F 511 (75.5%) 150 (22.2%) 16 (2.4%) 297 (43.9%) 351 (51.8%) 29 (4.3%)	387 (79.0%) 92 (18.8%) 11 (2.2%) 330 (67.3%) 152 (31.0%) 8 (1.6%) 441 (90.0%) 37 (7.6%) 12 (2.4%) 103.0 °F 445 (90.8%) 43 (8.8%) 2 (0.4%) 255 (52.0%) 217 (44.3%) 18 (3.7%)	174 (71.9%) 63 (26.0%) 5 (2.1%) 142 (58.7%) 94 (38.8%) 6 (2.5%) 208 (86.0%) 30 (12.4%) 4 (1.7%) 102.9 °F 189 (78.1%) 48 (19.8%) 5 (2.1%) 114 (47.1%) 115 (47.5%) 13 (5.4%)	50 (64.1%) 27 (34.6%) 1 (1.3%) 33 (42.3%) 44 (56.4%) 1 (1.3%) 63 (80.8%) 11 (14.1%) 4 (5.1%) 102.9 °F 47 (60.3%) 30 (38.5%) 1 (1.3%) 20 (25.6%) 56 (71.8%) 2 (2.6%)	186 (67.9%) 78 (28.5%) 10 (3.6%) 161 (58.8%) 101 (36.9%) 12 (4.4%) 240 (87.6%) 25 (9.1%) 9 (3.3%) 103.0 °F 247 (90.1%) 25 (9.1%) 2 (0.7%) 176 (64.2%) 92 (33.6%) 6 (2.2%)	4960 (63.3%) 2565 (32.7%) 311 (4.0%) 752 (9.6%) 4106 (52.4%) 4960 (63.3%) 5359 (68.4%) 1691 (21.6%) 786 (10.0%) 102.8 °F 4916 (62.7%) 2633 (33.6%) 287 (3.7%) 3030 (38.7%) 4496 (57.4%) 310 (4.0%)
**Vaccination** EHV-1/-4 yes EHV-1/-4 no EHV-1/-4 unknown EIV yes EIV no EIV unknown *S. equi* yes *S. equi* no *S. equi* unknown	3394 (33.0%) 965 (9.4%) 5937 (57.7%) 3344 (32.5%) 921 (8.9%) 6031 (58.6%) 988 (9.6%) 3224 (31.3%) 6084 (59.1%)	158 (22.6%) 108 (15.5%) 433 (61.9%) 153 (21.9%) 102 (14.6%) 444 (63.5%) 32 (4.6%) 201 (28.8%) 466 (66.7%)	220 (32.5%) 50 (7.4%) 407 (60.1%) 221 (32.6%) 47 (6.9%) 409 (60.4%) 55 (8.1%) 198 (29.2%) 424 (62.6%)	156 (31.8%) 35 (7.1%) 299 (61.0%) 152 (31.0%) 33 (6.7%) 305 (62.2%) 50 (10.2%) 125 (25.5%) 315 (64.3%)	69 (28.5%) 30 (12.4%) 143 (59.1%) 63 (26.0%) 32 (13.2%) 147 (60.7%) 21 (8.7%) 73 (30.2%) 148 (61.2%)	31 (39.7%) 3 (3.8%) 44 (56.4%) 33 (42.3%) 4 (5.1%) 41 (52.6%) 7 (9.0%) 25 (32.1%) 46 (59.0%)	63 (23.0%) 35 (12.8%) 176 (64.2%) 66 (24.1%) 33 (12.0%) 175 (63.9%) 25 (9.1%) 67 (24.5%) 182 (66.4%)	2697 (34.4%) 704 (9.0%) 4435 (56.6%) 2656 (33.9%) 670 (8.6%) 4510 (57.6%) 798 (10.2%) 2535 (32.4%) 4503 (57.5%)

**Table 4 pathogens-11-00759-t004:** Multiple-pathogen detection in 274 horses with acute onset of fever and/or respiratory signs participating in a voluntary surveillance program for infectious respiratory pathogens.

qPCR Positive Co-Infections	Number of Horses (%)
EIV + EHV-4 EIV + *S.equi* EIV + ERVs EIV + EHV-1 EHV-4 + *S.equi* EHV-4 + ERVs EHV-4 + EHV-1 *S.equi* + ERVs *S.equi* + EHV-1 ERVs + EHV-1 EIV + EHV-4 + EHV-1 EIV + EHV-4 + *S. equi* EIV + EHV-4 + ERVs EIV + *S. equi* + ERVs EHV-4 + *S. equi* + ERVs EHV-4 + EHV-1 + *S. equi* EHV-1 + *S. equi* + ERVs	53 (19.3%) 25 (9.1%) 18 (6.6%) 1 (0.3%) 51 (18.6%) 35 (12.8%) 21 (7.7%) 48 (17.5%) 5 (1.8%) 2 (0.7%) 2 (0.7%) 3 (1.1%) 1 (0.3%) 2 (0.7%) 4 (1.5%) 1 (0.3%) 2 (0.7%)

## Data Availability

Data available on request due to privacy restrictions.

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
