# Peer review of "Frequency of Detection and Prevalence Factors Associated with Common Respiratory Pathogens in Equids with Acute Onset of Fever and/or Respiratory Signs (2008–2021)"

_pathogens, 2022, doi:10.3390/pathogens11070759_

Round 1

Reviewer 1 Report

This is a very nice study that has been done that focuses on pathogen diagnosis in horses with fever and/or signs of respiratory tract disease.  It would add to clarity of this paper on how the participating practices were choosing to participate in this , or in an alternative study. I.e. , how can we make sure a representative subset of samples was sent via this sample set acquisition and select samples were redirected towards ‘other’ laboratories.

About 70% of cases were undiagnosed. Could the authors please elaborate on concrete ‘other’ pathogens that may have been missed in this study. Also in those cases with ‘fever only’?

Fevers during EHV-1 infections in an adult horse population is likely due to viremia.  Nasal swab samples can be false negative in case of an EHV-1 infection. I think the author should go into greater detail to discuss the 70% discrepancies.

The serum tubes were or were not analyzed? And if so, for what?

Few stumble blocks:

L 210: 2x the focus on EI and EHV-4 for seasonality?

L227: The demographic of the two lesser respiratory viruses, ERVs and EHV-1, showed similar prevalence factors as EHV-4 and EIV, respectively.

L232 On the other hand, EHV-1 qPCR-positive study horses presented predominantly during the spring months and were often older in age. How similar is similar in line with L227?

L237: Unfortunately, the outcome of EHV-1 qPCR-positive study cases and possible EHM outbreaks was not documented in the present study.

Author Response

This is a very nice study that has been done that focuses on pathogen diagnosis in horses with fever and/or signs of respiratory tract disease.  It would add to clarity of this paper on how the participating practices were choosing to participate in this , or in an alternative study. I.e. , how can we make sure a representative subset of samples was sent via this sample set acquisition and select samples were redirected towards ‘other’ laboratories.

The reviewer brings up a very relevant point, which is the bias selection of enrolled practices. The voluntary biosurveillance program enrolled equine practices from the client directory of Merck Animal Health. Merck Animal Health is one of the three large veterinary pharmaceutical company represented in the USA. While the enrollment was not random, it included practices from 44 US regions with number of practices reflecting horse distributions (37 practices from the eastern US regions, 45 practices from midwestern regions, 82 practices from southern regions and 97 practices from western regions). The mentioned limitation was listed in the discussion.

About 70% of cases were undiagnosed. Could the authors please elaborate on concrete ‘other’ pathogens that may have been missed in this study. Also in those cases with ‘fever only’?

There are various possible explanations for the relatively low yield in well characterized pathogens, including case selection (respiratory signs without fever), stage of disease (chronic versus acute), sample collected (rostral nasal secretions versus deeper pharyngeal samples) and pathogen detected. The study focused on major respiratory pathogens (EIV, EHV-1/-4, ERAV, ERBV and S. equi ss equi). Other less-characterized pathogens such as adenovirus, EHV-2, EHV-5 and S. equi ss zooepidemicus have been occasionally associated with respiratory disease. The difficulty in determining association between less-characterized pathogens and respiratory disease is their ubiquitous nature. Therefore, the lesser characterized pathogens were not tested in this study. In recent years, a variety of novel equine viruses (parvoviruses, picornavirus, protoparvovirus, and copivirus) have been discovered in respiratory secretions of horses with respiratory disease. However, no causality has yet been determined for these novel viruses. The names of the novel viruses have been added in the discussion.

Fevers during EHV-1 infections in an adult horse population is likely due to viremia.  Nasal swab samples can be false negative in case of an EHV-1 infection. I think the author should go into greater detail to discuss the 70% discrepancies.

While it is true that the second febrile peak of EHV-1 infection is due to viremia, most horses are presented to equine veterinarians during the first febrile peak, which represents the infection of the upper airways. Unfortunately, blood was not collected from the index cases. In the authors’ opinion, the value of blood would have been in performing serology as previous epidemiological studies have shown that the detection of recent infection associated with respiratory disease is higher when serology is used rather than antigen detection (culture and/or qPCR). The latter point is likely the reason why the overall detection rate is low. The authors have added information pertaining to this relevant topic in the discussion and have listed reasons for the discrepancy under limitations.

The serum tubes were or were not analyzed? And if so, for what?

This information was listed by mistake and was removed.

Few stumble blocks:

L 210: 2x the focus on EI and EHV-4 for seasonality?

The sentence was reworded to prevent any confusion.

L227: The demographic of the two lesser respiratory viruses, ERVs and EHV-1, showed similar prevalence factors as EHV-4 and EIV, respectively.

The sentence was reworded to prevent any confusion.

L232 On the other hand, EHV-1 qPCR-positive study horses presented predominantly during the spring months and were often older in age. How similar is similar in line with L227?

The sentence was reworded to prevent any confusion.

L237: Unfortunately, the outcome of EHV-1 qPCR-positive study cases and possible EHM outbreaks was not documented in the present study.

EHM is a complication of EHV-1 infection and sporadically occurs following the development of respiratory signs. The study designs did not allow us to follow up on EHV-1 qPCR-positive horses and the development of further complications. To avoid any confusion, the sentence was removed from the discussion.

Reviewer 2 Report

General comments

The authors present information on the frequency of identification of common pathogens in acute respiratory disease of horses, and identify factors associated with the individual pathogens. The information is interesting and adds to the available literature. The manuscript is well written; the data presentation could be improved if significant findings could be marked/indicated in the tables.

The authors state in their methods that “Case inclusion was restricted to horses with acute onset of disease and multiple horse submissions (up to 5 clinical cases per premise) were encouraged for instances where an outbreak was suspected.” In the results section and discussion, the authors repeatedly refer to the evaluation of “index cases”. If multiple samples (multiple horses) from a premise were collected, were all samples/horses included in the analysis? If so, the discussion of only index cases seems unjustified. If not - how was the “index case” defined, and how were additional samples from the same premise handled in the analysis? Including multiple samples from an outbreak would affect the independence of data points, which would need to be considered in the analysis.

The authors state in their methods, that only horses with a single pathogen were included in the analysis. In the results, however, they present a statistical comparison between horses with single pathogens to those with multiple pathogens. Please include this comparison in the statistics section.  

Specific comments

Line 52: on their epidemiology

Table 3: temperature is misspelled (temperture)

Table 3: under “vaccination EIV”, “unknown” is misspelled (unknwon)

Table 4 title: “Co-infections associated with 274 horses” sound unusual, please consider re-phrasing

Line 93: “group” is missing after “5-9 years of age”

Line 103: Please give accurate P-values for lethargy and anorexia, or explain why these are not considered necessary. The same comment in principle goes for line 118, line 126, line 130, line 141, line 152, line 154

Line 122: What is the definition of “older horses” in this context? Which age categories in table 2 does this refer to?

Line 139: Is this correct (ERVs qPCR-positive cases were less commonly associated with ….. single affected horses on the property (P < 0.001)) – it appears that the percentage of single cases is quite high?

Line 168: Can a more accurate P-value than P < 0.009 be given?

Line 212: Should this read “the reason for differences in seasonal detection”?

Line 216: This sentence is a bit misleading, as a statistical difference was only detected for the age category 5-9 years of age. Please revise.

Line 220/221: Please rephrase (“The high frequency of EIV qPCR- positive horses affecting multiple horses on a premise …”)

Line 233: This sentence is a bit misleading as EHV-1 infections were also more common in the age category 1-4 years. Please revise.

Line 234: The results section does not contain information that EHV-1 was more frequently identified in horses used for competition, please add.

Line 242: Also see the comment to line 122 – which age category (or categories) does the “older horses” refer to? According to line 266, infection with S. equi equi was more common in horses > 5 years of age? In the discussion, please differentiate between statistically significant results and descriptive results.

Line 273: the study data do not offer (data is plural)

Line 351/352: As fever and respiratory signs (cough or nasal discharge) were inclusion criteria for the study, this is a circular conclusion and the sentence (“Common clinical signs….”) should be removed.

Author Response

General comments

The authors present information on the frequency of identification of common pathogens in acute respiratory disease of horses, and identify factors associated with the individual pathogens. The information is interesting and adds to the available literature. The manuscript is well written; the data presentation could be improved if significant findings could be marked/indicated in the tables.

The authors state in their methods that “Case inclusion was restricted to horses with acute onset of disease and multiple horse submissions (up to 5 clinical cases per premise) were encouraged for instances where an outbreak was suspected.” In the results section and discussion, the authors repeatedly refer to the evaluation of “index cases”. If multiple samples (multiple horses) from a premise were collected, were all samples/horses included in the analysis? If so, the discussion of only index cases seems unjustified. If not - how was the “index case” defined, and how were additional samples from the same premise handled in the analysis? Including multiple samples from an outbreak would affect the independence of data points, which would need to be considered in the analysis.

As mentioned by the reviewer, the study enrolment for each clinic was restricted to horses with acute onset of fever and/or respiratory signs. Multiple submissions were encouraged. The authors have reviewed all the submissions and multiple cases from the same premise were not submitted. To prevent any confusion, the statement of “up to 5 clinical cases per premise” was removed.

The authors state in their methods, that only horses with a single pathogen were included in the analysis. In the results, however, they present a statistical comparison between horses with single pathogens to those with multiple pathogens. Please include this comparison in the statistics section. 

To avoid interpretation bias when multiple pathogens were involved, only horses with a single pathogen were evaluated in each specific group. Horses with co-infections were grouped in a multiple pathogen group. This additional information was added in the material and methods.

Specific comments

Line 52: on their epidemiology

The wording was corrected.

Table 3: temperature is misspelled (temperture)

Temperature was spelled correctly.

Table 3: under “vaccination EIV”, “unknown” is misspelled (unknwon)

Unknown was spelled correctly.

Table 4 title: “Co-infections associated with 274 horses” sound unusual, please consider re-phrasing

The title of Table 4 was changed as suggested

Line 93: “group” is missing after “5-9 years of age”

The missing word was added.

Line 103: Please give accurate P-values for lethargy and anorexia, or explain why these are not considered necessary. The same comment in principle goes for line 118, line 126, line 130, line 141, line 152, line 154

Non-significant results were listed as P value > 0.05 according to the Journal’s past recommendations.

Line 122: What is the definition of “older horses” in this context? Which age categories in table 2 does this refer to?

Older refers to horses older than 1 year of age. The age group was added in the results.

Line 139: Is this correct (ERVs qPCR-positive cases were less commonly associated with ….. single affected horses on the property (P < 0.001)) – it appears that the percentage of single cases is quite high?

Yes, the reviewer is right, there were 162 ERVs qPCR-positive horses (66.9%) as single affected individuals, versus 59 (24.4%) of ERVs qPCR-positive horses from farms with multiple affected horses.

Line 168: Can a more accurate P-value than P < 0.009 be given?

The value was changed to P < 0.01 to be consistent with the reporting of values in the results.

Line 212: Should this read “the reason for differences in seasonal detection”?

Thank you for the suggested changes, which were incorporated in the discussion.

Line 216: This sentence is a bit misleading, as a statistical difference was only detected for the age category 5-9 years of age. Please revise.

The changes were mad according to the data.

Line 220/221: Please rephrase (“The high frequency of EIV qPCR- positive horses affecting multiple horses on a premise …”)

The sentence was changed as suggested by the reviewer.

Line 233: This sentence is a bit misleading as EHV-1 infections were also more common in the age category 1-4 years. Please revise.

The sentence was changed to reflect both age groups with high frequency of EHV-1 infection (1-4 and 10-14 years).

Line 234: The results section does not contain information that EHV-1 was more frequently identified in horses used for competition, please add.

The missing information was added as requested by the reviewer.

Line 242: Also see the comment to line 122 – which age category (or categories) does the “older horses” refer to? According to line 266, infection with S. equi equi was more common in horses > 5 years of age? In the discussion, please differentiate between statistically significant results and descriptive results.

The statement applies to horses older than 1 year of age. The sentence on line 1222 and 212 was changed.

Line 273: the study data do not offer (data is plural)

Thank you for catching this spelling mistake, which was corrected.

Line 351/352: As fever and respiratory signs (cough or nasal discharge) were inclusion criteria for the study, this is a circular conclusion and the sentence (“Common clinical signs….”) should be removed.

The sentence has been changed to reflect that the distribution of clinical signs was similar amongst the various infection groups. 

Round 2

Reviewer 1 Report

Dear Authors,

After reading the revised version- EHV-1 diagnosis has been made only on nasal swabs, and not on EDTA blood buffy coat. The paper needs a comment that EHV-1 data could be underestimated. 

Also,  as EHV-1 and EHM in particular are notifiable diseases in many US states a sampling bias is likely to occur with the sample submission process. Again, this could underestimate the EHV-1 data set and needs to be added to the manuscript's discussion. 

Author Response

After reading the revised version- EHV-1 diagnosis has been made only on nasal swabs, and not on EDTA blood buffy coat. The paper needs a comment that EHV-1 data could be underestimated. 

The authors thank the reviewer for pointing out that blood was not tested for EHV-1. The authors agree that in some instances, although rare, horses may be viremic in the absence of viral shedding via the nasal passages. This situation may lead to an underrepresentation of true EHV-1 respiratory cases. The discussion was expanded to include this limitation.

Also,  as EHV-1 and EHM in particular are notifiable diseases in many US states a sampling bias is likely to occur with the sample submission process. Again, this could underestimate the EHV-1 data set and needs to be added to the manuscript's discussion. 

While EHM is a reportable disease in many US states, rhinopneumonitis is not. The goal of the biosurveillance study was to focus on respiratory infection and not on complications of specific pathogens such as EHM or even abortion. Therefore, the authors do not believe that state mandated regulation on EHM would have negatively impacted or biased case selection and case submission.